# Production of 4-Deoxy-L-*erythro*-5-Hexoseulose Uronic Acid Using Two Free and Immobilized Alginate Lyases from *Falsirhodobacter* sp. Alg1

**DOI:** 10.3390/molecules27103308

**Published:** 2022-05-21

**Authors:** Yuzuki Tanaka, Yoshihiro Murase, Toshiyuki Shibata, Reiji Tanaka, Tetsushi Mori, Hideo Miyake

**Affiliations:** 1Graduate School of Bioresources, Mie University, 1577 Kurimamachiya-cho, Tsu 514-8507, Mie, Japan; 521m308@m.mie-u.ac.jp (Y.T.); xxx.3569ys@gmail.com (Y.M.); shibata@bio.mie-u.ac.jp (T.S.); tanakar@bio.mie-u.ac.jp (R.T.); 2Seaweed Biorefinery Research Center, Mie University, 1577 Kurimamachiya-cho, Tsu 514-8507, Mie, Japan; 3Department of Biotechnology and Life Science, Tokyo University of Agriculture and Technology, Koganei 184-8588, Tokyo, Japan; moritets@go.tuat.ac.jp

**Keywords:** alginate lyase, 4-deoxy-L-*erythro*-5-hexoseulose uronic acid (DEH), *Falsirhodobacter* sp. alg1, immobilized enzyme

## Abstract

*Falsirhodobacter* sp. alg1 expresses two alginate lyases, AlyFRA and AlyFRB, to produce the linear monosaccharide 4-deoxy-L-*erythro*-5-hexoseulose uronic acid (DEH) from alginate, metabolizing it to pyruvate. In this study, we prepared recombinant AlyFRA and AlyFRB and their immobilized enzymes and investigated DEH production. Purified AlyFRA and AlyFRB reacted with sodium alginate and yielded approximately 96.8% DEH. Immobilized AlyFRA and AlyFRB were prepared using each crude enzyme solution and *κ*-carrageenan, and immobilized enzyme reuse in batch reactions and DEH yield were examined. Thus, DEH was produced in a relatively high yield of 79.6%, even after the immobilized enzyme was reused seven times. This method can produce DEH efficiently and at a low cost and can be used to mass produce the next generation of biofuels using brown algae.

## 1. Introduction

Renewable energy using biomass based on carbon neutrality through the carbon cycle is emerging to reduce carbon dioxide emissions in the environment. Biomass can be terrestrial or aquatic. Aquatic biomass has several advantages in biorefinery: (1) it does not require fertilizer for its growth; (2) it does not compete with other food crops; and (3) it is easily saccharified by enzymes because it does not contain lignin, which is contained in terrestrial biomass. Seaweeds are mainly brown, red, and green algae. According to the Food and Agriculture Organization of the United Nations (FAO), brown algae accounts for 76.6% of the world’s wild seaweed collections [1]. Global seaweed aquaculture production tonnage, mostly brown and red algae, increased 1000-fold from 34.7 thousand tons to 34.7 million tons between 1950 and 2019 [1]. Many brown algae have large, well-developed bodies composed of polysaccharides, including alginate, fucoidan, cellulose, hemicellulose, and laminarin. Of which, alginate is the most abundant, accounting for approximately 40% depending on the type of brown algae [2]. Alginate, an intercellular polysaccharide, is a linear polysaccharide consisting of two types of uronic acids, β-D-mannuronate and α-L-guluronate, linked by 1,4-glycosidic bonds.

Alginate-utilizing microorganisms express endo- and exo-type alginate lyases, degrading alginate to unsaturated uronic acid monosaccharides. Subsequently, its pyranose ring is enzymatically or non-enzymatically cleaved to produce 4-deoxy-L-*erythro*-5-hexoseulose uronic acid (DEH) [3,4,5]. We isolated *Falsirhodobacter* sp. alg1 having high alginate-degrading activity, by screening microorganisms that assimilate brown algae [6]. Genome analysis revealed that its genome encodes an endo-type (*alyFRA*) and an exo-type (*alyFRB*) alginate lyase gene, and their amino acid sequence information indicated that they belong to the polysaccharide lyase family (PL) 7 (PL7) and PL15, respectively [7]. Recently, strains of *Escherichia coli*, *Saccharomyces cerevisiae*, and *Sphingomonas* sp. A1 were created by introducing alginate lyase and other enzyme genes necessary to metabolize DEH and produce bioethanol [8,9,10,11,12,13]. Although these strains can produce ethanol directly from sodium alginate, they cannot use high concentrations of sodium alginate in the medium because of its high viscosity. However, DEH, a monosaccharide, is less viscous and can be treated like glucose. In 2014, Wang et al. used the recombinant enzymes, endo-alginate lyase Alg7D and exo-alginate lyase Alg17C, from the marine bacterium *Saccharophagus degradans,* to obtain a 45.5% DEH yield (DEH weight/alginate weight) from alginic acid [14]. For industrial applications, increasing the yield of DEH and considering enzyme reuse to minimize the cost is necessary. Enzyme reuse can generally be overcome by immobilized enzymes. In addition, because the immobilized enzymes can be handled as solids, separating them from the reaction solution containing the products can be easy, minimizing the contamination of proteins in the reaction solution. Optimizing the method for preparing immobilized enzymes and reaction conditions improves the number of immobilized enzyme reuses, enabling lower costs [15]. In this study, we investigated DEH production using free and immobilized alginate lyases, AlyFRA and AlyFRB, from *Falsirhodobacter* sp. alg1.

## 2. Results

### 2.1. Preparation of Crude and Recombinant AlyFRA and AlyFRB

The *alyFRA* or *alyFRB* genes were incorporated into pET vectors and transformed into *E. coli* BL21 (DE3) and those transformants were cultured. Each cell was disrupted by ultrasonication. Most of the expressed AlyFRA and AlyFRB were soluble, which were used as crude enzyme solutions for the immobilized enzyme experiments. The enzymes were then purified using Ni affinity chromatography to remove the attached His-tag. As a result, a single recombinant AyFRA (rAyFRA) and AlyFRB (rAyFRB) were obtained (Figure 1). Theoretical molecular weights calculated from the primary sequences of rAlyFRA and rAlyFRB were 89 and 52 kDa, respectively, consistent with the molecular weights calculated from the band positions shown on sodium dodecyl sulfate-polyacrylamide gel electrophoresis (SDS-PAGE). Each enzyme yielded 4.7 mg rAlyFRA and 10.0 mg rAlyFRB per liter of medium.

### 2.2. Enzyme Reaction Products

The reaction products of rAlyFRA or rAlyFRB using sodium alginate as a substrate were detected using thin-layer chromatography (TLC) (Figure 2a). When only rAlyFRA was used, alginate oligosaccharides of different degrees of polymerization were obtained. When only rAlyFRB or both rAlyFRA and rAlyFRB were used, a single spot was observed at the same position as glucose; the spot from both rAlyFRA and rAlyFRB was larger than that from only rAlyFRB. The reaction product of both rAlyFRA and rAlyFRB was analyzed using liquid chromatography-mass spectrometry (LC/MS), and one peak was detected in the total ion current (TIC) chromatogram (Figure 2b). The peak detected in the selected ion monitoring (SIM) chromatogram with a mass/charge ratio (*m*/*z*) of 175 (indicating the mass number of deprotonated DEH) was consistent with retention time in the TIC (Figure 2c). These results indicate that pure DEH is obtained when both rAlyFRA and rAlyFRB are used. The yields of DEH when rAlyFRA and rAlyFRB were used are shown in Table 1. When rAlyFRA and rAlyFRB reacted at a concentration of 1:2, DEH was obtained in 96.7% yield (DEH weight/alginate weight).

### 2.3. Immobilization Rate of AlyFRA and AlyFRB

Various immobilization matrices, including agar, alginate, pectin, polyacrylamide, and carrageenan, have been used for preparing immobilized enzymes [16]. The immobilized enzymes were prepared using 1.5–4.0% (*w*/*v*) *κ*-carrageenan and crude enzymes AlyFRA or AlyFRB (Figure 3). The immobilization rates of these enzymes are shown in Figure 4. Immobilized AlyFRA (iAlyFRA) or AlyFRB (iAlyFRB) leaked 10.6–15.2% or 6.2–11.9% protein, respectively, while being soaked in the KCl solution during immobilization. However, after 7 d of soaking iAlyFRA or iAlyFRB in 20 mM phosphate buffer (pH 7.4) containing 100 mM NaCl from KCl solution, the immobilization rate of iAlyFRA or iAlyFRB was 84.8–89.4% or 84.1–90.6%, respectively.

### 2.4. Batch Reaction of Immobilized Enzymes

Seven batch reactions were performed in which iAlyFRA and iAlyFRB immobilized with 2.0%, 3.0%, and 4.0% (*w*/*v*) *κ*-carrageenan were reacted with 1.0% (*w*/*v*) low molecular weight sodium alginate as a substrate. The immobilized enzymes were extracted from the reaction solution and reacted with a new substrate daily. Each reaction solution was examined for products using TLC (Figure 5). Some small spots corresponding to alginate oligosaccharides were observed, but all reaction solutions showed spots in the same position as DEH throughout the seven batch reactions. However, a substrate spot was detected in the third and sixth batch reactions for the enzymes immobilized with 2.0% and 3.0% *κ*-carrageenan (*w*/*v*), respectively, at the position where the reaction solution was applied. For those with 4% (*w*/*v*) *κ*-carrageenan, all substrates were degraded in all batch reactions, and their main product was DEH. The average yield of DEH in all batch reactions with the enzymes immobilized with 4.0% *κ*-carrageenan was 79.6%.

## 3. Discussion

TLC and LC/MS analysis showed that the reaction of both rAlyFRA and rAlyFRB with sodium alginate produced highly purified DEH (Figure 2). DEH yield of approximately 96.7% (DEH weight/alginate weight) was obtained (Table 1), which was more than twice as high as the maximum yield of 45.5% when using Alg7D and Alg17C [14]. Many alginate-utilizing microorganisms have multiple conserved endo- and exo-type alginate lyase genes in their genomes, expressed as proteins to degrade alginate. However, *Falsirhodobacter* sp. alg1 possesses a single endo- and exo-type alginate lyase gene conserved in its genome, making it a unique alginate-utilizing microbe [6,7]. Therefore, *Falsirhodobacter* sp. alg1 degrades and metabolizes alginate in various brown algae using only two alginate lyases, AlyFRA and AlyFRB, efficiently producing DEH. Thus, we believe that using rAlyFRA and rAlyFRB can provide a high DEH yield of high purity.

AlyFRA- and AlyFRB-immobilized enzymes with *κ*-carrageenan were prepared using crude enzyme solutions (Figure 3), which can minimize the enzyme preparation cost. The immobilization rate of iAlyFRA and iAlyFRB increased with increasing *κ*-carrageenan concentration (Figure 4). In particular, the immobilization rate remained above 85% when *κ*-carrageenan concentration was above 2.5% (*w*/*v*). The *κ*-carrageenan gel became firmer and less fragile. Furthermore, the immobilized enzymes prepared with 4.0% (*w*/*v*) *κ*-carrageenan could completely degrade the substrate and produce DEH in the seventh batch reaction (Figure 5). Therefore, immobilized enzymes with 4.0% (*w*/*v*) *κ*-carrageenan can be reused several times, providing approximately 80% DEH yield. This method can produce large amounts of DEH from sodium alginate at a low cost. The industrial production of sodium alginate is well established but environmentally hazardous due to using acids and alkalis. As a future challenge, a method to extract saccharified liquid containing alginic acid from brown algae using microorganisms with a low environmental impact is required.

In our previous study, we successfully created a *Saccharomyces cerevisiae* strain capable of metabolizing sodium alginate and producing ethanol by introducing the enzyme genes required for alginate lyase and DEH metabolism into *S. cerevisiae* [11,12]. In this study, we demonstrated a method to produce large quantities of DEH from sodium alginate at a low cost, which is expected to lead to the research and development of biofuels using large seaweeds.

## 4. Materials and Methods

### 4.1. Overexpression and Purification of Recombinant AlyFRA and AlyFRB

The previously prepared expression plasmids of AlyFRA and AlyFRB for *E. coli* were used [7]. The expression plasmid of AlyFRB was modified by cloning polymerase chain reaction-amplified *alyFRB* open reading frame into the pET22b(+) vector at the *Nde*I and *Xho*I sites using standard cloning techniques to increase the expression level. Each plasmid DNA was transformed into *E. coli* BL21 (DE3). A transformant was grown in a 2× YT medium (1.6% Hipolypepton [FUJIFILM Wako Pure Chemical Corporation, Osaka, Japan], 1.0% yeast extract [Nacalai Tesque, Inc., Kyoto, Japan], and 0.5% NaCl) containing 1.0% glucose and 100 μm ampicillin at 37 °C to an optical density of 0.6–0.8 at 600 nm, induced with isopropyl β-D-thiogalactopyranoside (0.1 mM final concentration), and harvested by centrifugation at 20,000× *g* after 40 h at 20 °C. Bacteria were harvested and suspended in 20 mM sodium phosphate buffer (pH 7.4) containing 100 mM NaCl. The cells were disrupted by ultrasonication, and the soluble fraction of the mixture was collected by centrifugation at 20,000× *g* for 20 min at 4 °C. The cell-free extract was used as a crude enzyme solution of AlyFRA and AlyFRB.

Each crude enzyme solution was applied to a Ni Sepharose 6 Fast Flow column (GE Healthcare, Chicago, IL, USA) equilibrated with 20 mM sodium phosphate buffer (pH 7.4) containing 20 mM imidazole and 500 mM NaCl. After washing the column with the same buffer, 20 mM sodium phosphate buffer (pH 7.4) containing 500 mM imidazole and 500 mM NaCl was eluted. The fractions were dialyzed against 100 mM Tris-HCl buffer (pH 7.5) containing 100 mM NaCl. The amount of each recombinant protein obtained at each purification step was monitored using SDS-PAGE. The purified protein was found to be >95% pure according to SDS-PAGE analysis (Figure 1). Each recombinant protein was used with the His-tag attached. The protein concentration of the crude enzyme solution was calculated using Quick Start^TM^ Protein Assay (Bio-Rad Laboratories, Inc., Hercules, CA, USA) based on the Bradford method. The purified protein concentration was measured spectrophotometrically using the molar attenuation coefficient of those proteins at 280 nm [17].

### 4.2. TLC Analysis of Products

Sodium alginate (Sigma-Aldrich, St. Louis, MO, USA) was used as the substrate. To identify the products of rAlyFRA or rAlyFRB, 0.2 mg of purified AlyFRA or AlyFRB and 0.1 g of the substrate were incubated in 10 mL of 10 mM Tris-HCl buffer (pH 7.5) at 25 °C for 3 d. The products were analyzed using TLC (glass TLC plate, silica gel 60; Merck, Darmstadt, Germany). The TLC plate sample was prepared using a mixture of butanol, acetic acid, and water in a 2:1:1 ratio by volume. After development, the spots of sugars were visualized by heating the TLC plate sprayed with diphenylamine–aniline–phosphoric acid (DPA) reagent. DPA regent was prepared using the method described by Bailey et al. [18].

### 4.3. Identification of Products by LC/MS Analysis

DEH was separated and detected using 6120 Quadrupole LC/MS with a 1260 Series HPLC system comprising variable wavelength detectors, quaternary pumps, autosampler, and thermostatted column compartments (Agilent Technologies, Santa Clara, CA, USA). The UV detector was set at 235 nm. TIC profile and SIM analysis were performed by a previously described method [19].

### 4.4. Product Yield

rAlyFRA and rAlyFRB were used as free enzymes. rAlyFRA and rAlyFRB at 0.2–0.6 mg/mL concentration were reacted with 1.0% (*w*/*v*) sodium alginate as a substrate and 20 mM Tris-HCl buffer (pH 7.5) for 3 d at 25 °C. Ultrafiltration was performed using an Amicon Ultra-15 3000 MWCO spin concentrator (Merck) to remove unreacted sodium alginate and the enzymes. The reaction solution passed through the ultrafiltration filter and was freeze-dried to obtain a product. The net product weight was defined as the product weight after lyophilization minus the solute weight in the buffer. The product yield was indicated as the product weight/sodium alginate weight. Each experiment was performed three times.

### 4.5. Preparation of Immobilized Enzyme

Several 17 mL vials of 20 mM phosphate buffer (pH 7.4) containing 100 mM NaCl were prepared. *κ*-carrageenan (Tokyo Chemical Industry Co., Ltd., Tokyo, Japan) was added to each buffer solution to obtain concentrations of 1.5%, 2.0%, 2.5%, 3.0%, 3.5%, and 4.0% (*w*/*w*) to prepare 20 mL reaction solutions. After dissolving *κ*-carrageenan in a hot water bath at 68–70 °C, the temperature was decreased to 55 °C with constant stirring. Crude AlyFRA and AlyFRB at 9–11 mg/mL and 12–17 mg/mL concentrations, respectively, were pre-incubated at 45 °C. In each solution containing *κ*-carrageenan, 3 mL of crude enzyme solution of AlyFRA or AlyFRB was added, and the mixture was stirred for 1 min. The mixture was gelled by cooling with ice, soaked in 2.0% (*w*/*v*) KCl solution, and incubated overnight at 4 °C. Each gel was cut into cubes of approximately 5 mm^2^ and incubated in the same solution at 4 °C for 2 h. The cubes were washed twice with 20 mM phosphate buffer (pH 7.4) containing 100 mM NaCl and used in experiments as iAlyFRA or iAlyFRB.

### 4.6. Immobilization Rate of AlyFRA and AlyFRB

iAlyFRA or iAlyFRB prepared with different *κ*-carrageenan concentrations were soaked in 100 mL of 20 mM phosphate buffer (pH 7.4) containing 100 mM NaCl for 7 d at 20 °C, and 1.5 mL of the supernatants were collected at a certain time. In addition, 1.5 mL of the KCl solution used to prepare the immobilized enzymes was collected. The immobilization rate of each immobilized enzyme was calculated using the following equation [20]:$$\mathrm{Immobilization}\;\mathrm{yield}\;\left(\%\right)=\frac{C_{i}V_{i}-C_{s}V_{s}}{C_{i}V_{i}}\times 100$$
where *C_i_* is the initial crude enzyme concentration, *V_i_* is the initial crude enzyme volume, *C_s_* is the crude enzyme concentration leaked into the supernatant, and vs. is the volume of the crude enzyme leaked into the supernatant. The protein concentration was measured as described above. The immobilization rate of the immobilized enzyme prepared under various conditions was calculated three times, and their average was determined as the final immobilization rate.

### 4.7. Batch Reaction of Immobilized Enzymes

In the batch reaction, 20 mL of AlyFRA or AlyFRB each was immobilized with 2.0–4.0% (*w*/*v*) *κ*-carrageenan by the above method and was added to 100 mL of 1.0% (*w*/*v*) low molecular weight sodium alginate (Kaigen Pharma Co., Ltd., Osaka, Japan) substrate at 20 °C. The reaction solution was replaced with a new substrate solution of the same volume daily for 7 d. The products in each reaction solution were analyzed using TLC. For each reaction solution, unreacted sodium alginate and damaged immobilized enzyme were extracted using an ultrafiltration device (UHP-25K; Advantec MFS, Inc., Dublin, CA, USA) equipped with a filter with a fractional molecular weight of 10,000. The product was obtained by lyophilizing the solution through ultrafiltration. The product yield produced by the immobilized enzymes was calculated using the same method as free enzymes.

## 5. Conclusions

In this study, recombinant endo- and exo-type alginate lyases were prepared from *Falsirhodobacter* sp. alg1, and their free and immobilized enzymes were used to produce DEH from sodium alginate. These purified free enzymes could produce DEH with approximately 96.8% yield. The immobilized enzymes with *κ*-carrageenan could produce a relatively high yield of 79.6% DEH, even after being reused seven times. Since this method can produce DEH efficiently and at a low cost, it can be used for the mass production of next-generation biofuels using brown algae.

## 6. Patents

Part of this study includes the content of the PCT application (WO2017175694).

## Figures and Tables

**Figure 1 molecules-27-03308-f001:**
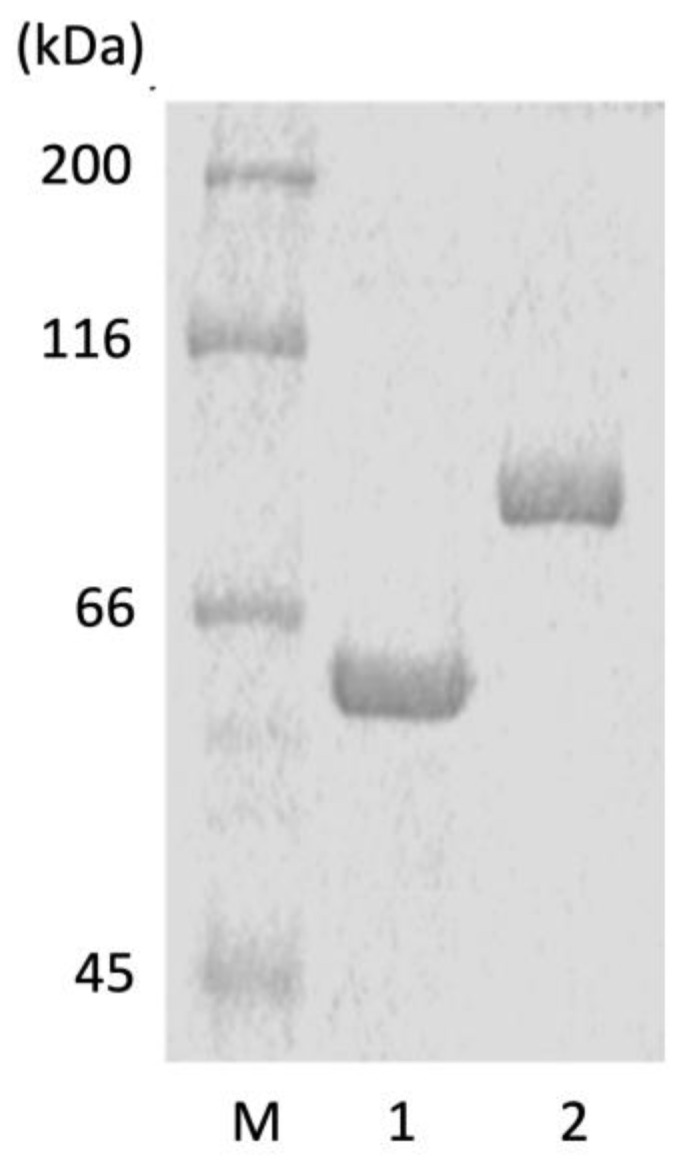
Sodium dodecyl sulfate–polyacrylamide gel electrophoresis (SDS-PAGE) analysis of the purified alginate lyases. SDS-PAGE was performed on 10% polyacrylamide gel. The gel was visualized by Coomassie brilliant blue. The samples were loaded in the following order: M, protein maker (molecular masses shown on the left); 1, rAlyFRA; and 2, rAlyFRB.

**Figure 2 molecules-27-03308-f002:**
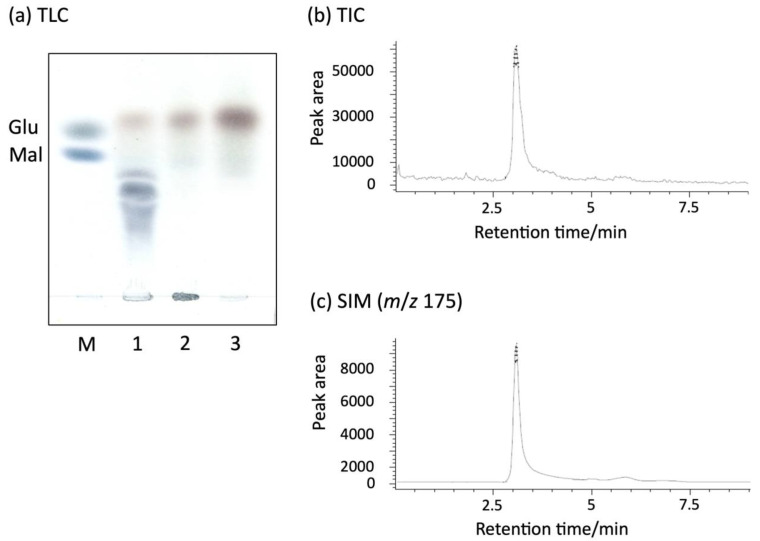
Degradation of sodium alginate by rAlyFRA or rAlyFRB. Sodium alginate reacted with rAlyFRA or rAlyFRB in 20 mM Tris-HCl buffer pH 7.5 at 25 °C for 3 d, and the products were analyzed using thin-layer chromatography (TLC) (**a**). M, Glucose and Maltose; 1, sodium alginate + AlyFRA; A2, sodium alginate + AlyFRB; and A3, sodium alginate + AlyFRA + AlyFRB. Identification of degraded products by AlyFRA + AlyFRB using total ion current (TIC) (**b**) and selected ion monitoring (SIM) (**c**) for identifying the degraded products of rAlyFRA and rAlyFRB. Peaks from both TIC and SIM at the retention time of 3 min indicate DEH.

**Figure 3 molecules-27-03308-f003:**
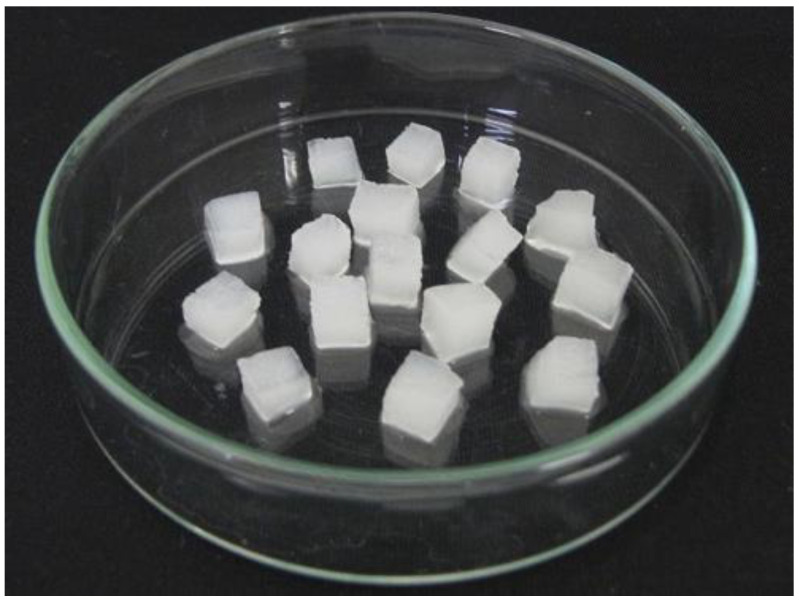
Immobilized enzyme prepared with *κ*-carrageenan. Immobilized enzymes were prepared using 1.5–4.0% (*w*/*v*) *κ*-carrageenan and crude enzymes AlyFRA or AlyFRB, which were cut into cubes of approximately 5 mm.

**Figure 4 molecules-27-03308-f004:**
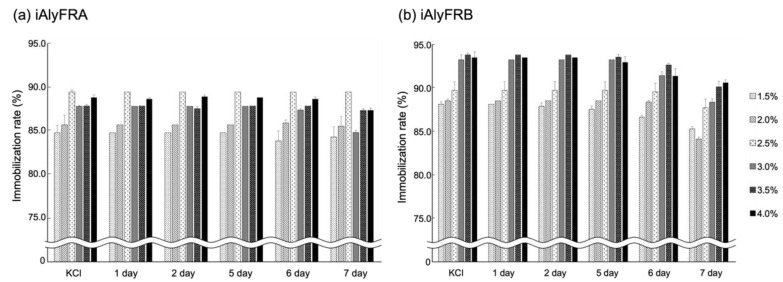
Immobilization rate of iAlyFRA (**a**) and iAlyFRB (**b**). iAlyFRA and iAlyFRB were prepared with 1.5–4.0% (*w*/*v*) *κ*-carrageenan soaked in 100 mL of 20 mM phosphate buffer (pH 7.4) containing 100 mM NaCl at 20 °C for 7 d. Each data point represents the average of the three independent experiments performed. Error bars show standard deviations (*n* = 3).

**Figure 5 molecules-27-03308-f005:**
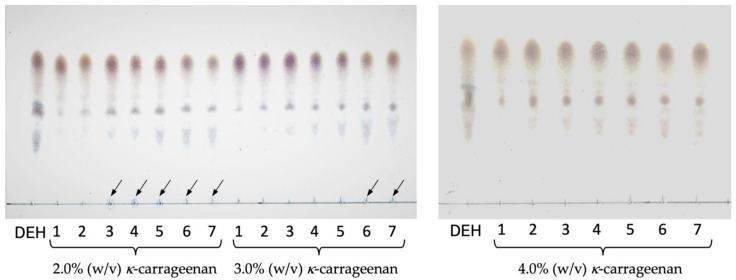
TLC analysis of products in batch reactions with immobilized enzymes. iAlyFRA and iAlyFRB were prepared with 2.0%, 3.0%, and 4.0% (*w*/*v*) *κ*-carrageenan. They were added to 1.0% (*w*/*v*) low molecular weight sodium alginate, and their reaction solutions were replaced daily. The numbers indicate the number of batch reactions. Arrows indicate undegraded substrates.

**Table 1 molecules-27-03308-t001:** DEH ^1^ yield.

rAlyFRA	rAlyFRB	Yield (%)
+ ^2^	+	84.8 ± 5.2
+	++	96.8 ± 1.2
+	+++	81.8 ± 3.0

^1^ Reaction conditions: rAlyFRA and rAlyFRB reacted with 1.0% (*w*/*v*) sodium alginate in 20 mM Tris-HCl buffer (pH 7.5) for 3 d at 25 °C. Each reaction was performed three times, and results are mean ± S.E.M. ^2^ Enzyme concentration: +, 0.20 mg/mL; ++, 0.40 mg/mL; +++, 0.60 mg/mL.

## Data Availability

Not applicable.

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
