# Peer review of "Production of 4-Deoxy-L-erythro-5-Hexoseulose Uronic Acid Using Two Free and Immobilized Alginate Lyases from Falsirhodobacter sp. Alg1"

_molecules, 2022, doi:10.3390/molecules27103308_

Round 1

Reviewer 1 Report

This manuscript describes the immobilization and evaluation of two alginate lyase enzymes derived from Falsirhodobacter.  There is great interest in conversion of biomass in to commodity chemicals or feedstocks, and this work focuses on this area, specifically the use of alginate derived from seaweed as a biomass source.  One the face of it, the results presented in the manuscript are very good: highly efficient conversion of alginate to a single ketoaldose using recyclable immobilized enzymes.  However, upon more careful reading one finds that the manuscript is written in such a way that it is impossible to tell exactly what the authors accomplished.  This is a combination of English language issues and scientific issues and must be addressed before the work is published in Molecules. 

It is impossible to tell exactly what proteins were immobilized.  The manuscript states the enzymes were purified by Ni-affinity chromatography "to remove the attached His-tag".  Does this mean the isolated proteins from expression in E. coli were subjected to enzymatic cleavage (what protease, what cleavage site) followed by Ni-column to remove the His-tag? Or is this just very poor wording for describing the purification of the His-tagged proteins?  But earlier in the paragraph, the authors indicate that "each cell" was disrupted by ultrasonication, that "most" of the expressed AlyFRA nd AlyFRB were soluble and "used as crude enzyme solution for immobilization".  So the question remains: was exactly did the authors do here? Did they immobilize the crude cell lysate from a heterologous expression of these enzymes in E. coli?  If this is the case, it is not clear that the observed activity on alginic acid can be attributed solely to the Aly enzymes activity.

What are the products of enzymatic degradation of alginic acid?  The analysis of the products by TLC and LC/MS have no standard to compare the afford mixtures to the claimed product, DEH.  The LC/MS analysis is not described in any detail, so this data would be impossible for someone else to reproduce. At a minimum, the column and elution conditions, as well as MS parameters must be provided.  The TIC chromatogram in Figure 2b looks considerably different from previously reported results, and the authors must provide a discussion of this difference.  

How much enzyme was immobilized?  The procedure described for quantifying the amount of enzyme immobilized is unclear, but appears NOT to follow the protocol described in the cited reference 20.

How efficient is the production of DEH?  The procedure described for quantifying the products of degradation of sodium alginate appear to simply measure the material that passes though a 3,000 Dalton cut-off spin filter, not pure DEH.  This must be addressed, as the major thrust of this manuscript is the highly efficient production of DEH, not simply the degradation of sodium alginate to lower molecular weight oligomers. This also holds true for the batch reactions with recycled immobilized enzymes.  The TLC analysis here clearly shows multiple spots, so the claim that DEH is the (only) product is misleading.

Author Response

We are grateful to reviewer #1 for their comments and useful suggestions that have helped us to improve our paper considerably. As indicated in the responses that follow, we have taken all these comments and suggestion into account while revising the manuscript.

Our manuscript has been edited in English by Editage. We experimented with recombinant enzymes with His-tag added. We did not remove His-tag from these because they had no effect on activity in previous study (Ref. 7). The manuscript also does not state that the added His-tag was removed. We have added the following sentences to the manuscript. Each recombinant protein was used with the His-tag attached. (Page 6, lines 204-205)

The free enzymes were purified using Ni columns. The enzymes used to prepare the immobilized enzymes were crude enzyme solutions. These are described in the Materials and Methods section.

When only rAlyFRA was used, the product were alginate oligosaccharides. When only rAlyFRB or both rAlyFRA and rAlyFRB were used, the product was DEH. These are also discussed in the manuscript.

There is no DEH as a reagent. The conditions for LC/MS are the same as for the method in Ref. 19, also described in the manuscript. The TIC chromatogram in Figure 2b showed a similar profile to Figure 2a in Ref. 19. Thus we believe that these results indicate that pure DEH is obtained when both rAlyFRA and rAlyFRB are used.

The immobilization rate is calculated as described in 4.6 Immobilization rate of AlyFRA and AlyFRB in Materials and Methods. (Ref. 20, 2.8.1 Immobilization yield in terms of protein.)

The average yield of DEH in all batch reactions with the enzymes immobilized with 4.0% κ-carrageenan was 79.6%. Ultrafiltration was performed to obtain pure DEH. Because the reaction solution contains unreacted sodium alginate and damaged immobilized enzyme. These are also discussed in the manuscript.

Small spots of alginate oligosaccharides were observed in the TLC analysis, but the main spot is DEH. The manuscript has been revised (Page 4, line 136).

Reviewer 2 Report

The only difference in the present article titled “Production of 4-deoxy-L-erythro-5-hexoseulose uronic acid 2 using two free and immobilized alginate lyases from 3 Falsirhodobacter sp. alg1” from the previous work ‘T. Mori et al. (PLOS ONE, 2016)’ is the immobilization of enzymes with carrageenan in the present study. Thus it is expected to bring out a clear novelty to the work. The article has a good flow of reading with the individual sections described clearly. A few more concerns need to be addressed before the article is considered for publication:

Introduction:

Although the word ‘renewable energy' is mentioned, the connection between DEH and renewable energy production is missing. What is meant by ‘renewable energy’? How DEH will contribute to ‘renewable energy’ production?

Discussion:

How much DEH yield is obtained without immobilizing the enzymes? Please elaborate with statistical significance and under the light of this work.  

Materials and Methods:

4.3 Please describe in brief the method for TIC profiling and SIM analysis.

4.5 The temperature selected for the addition of enzymes is 45°C. Does any degradation occur at this temperature? Does the variation in temperature have any effect on the final yield of DEH?

Line 179 and 279:

Please give a cost analysis detail as supplementary material or justification for the ‘low cost’ statement.

Author Response

We are grateful to reviewer #2 for their comments and useful suggestions that have helped us to improve our paper considerably. As indicated in the responses that follow, we have taken all these comments and suggestion into account while revising the manuscript.

In this study, recombinant endo- and exo-type alginate lyases were prepared from Falsirhodobacter sp. alg1, and their free and immobilized enzymes were used to produce DEH from sodium alginate and the yield of DEH was quantified.

Introduction:

Recently, strains of Escherichia coli, Saccharomyces cerevisiae, and Sphingomonas sp. A1 have been created by introducing alginate lyase and other enzyme genes necessary to metabolize DEH and produce bioethanol [Ref. 8–13]. In this study, we demonstrated a method to produce large quantities of DEH from sodium alginate, which is expected to lead to research and development of biofuels using large seaweeds.

Discussion:

The yields of DEH using the free enzyme are shown in Table 1. Each reaction was performed three times, and results are mean ± S.E.M.

Materials and Methods:

TIC profiles and SIM analysis were performed under the same conditions as in Ref. 19. The UV detector was set at 235 nm. The ionization method of the sample was tested via the ESI mode in negative mode. The peaks of DEH was detected using a mass spectrometer in SIM mode. In the SIM analysis, the m/z was set to 175 (corresponding to the molecular mass of the deprotonated form of DEH). Other experimental conditions for the mass spectrometer were as follows: dry gas, 12.0 L/min; nebulizer, 0.241 MPa; dry temperature, 250°C; vaporizer, 200°C; scan from m/z 100 to 1000. The column used in this research was Shodex IC NI-424 column (4.6 mm i.d. x 100 mm, Showa Denko). Elution was performed at a flow rate of 0.5 mL/min with a 40 mM ammonium formate buffer including 0.1% formic acid (pH 3.5) as a mobile phase. The column oven was set at 40°C.

To mix with heated k-carrageenan, the enzyme was set to 45°C. k-Carrageenan begins to solidify at temperatures below 50°C. The enzyme does not denature at this temperature, so the DEH yield is not affected.

Line 179 and 279:

It is not possible to calculate costs exactly. In general, enzyme preparations are the most costly when producing sugars from biomass. In this study, we believe that the use of immobilized enzymes will reduce costs because they can be reused at least seven times.

Reviewer 3 Report

The manuscript by Tanaka et al. presents the characterization of two alginate lyases from Falsirhodobacter sp. alg1, a bacterial strain isolated and studied by the same group. These two enzymes were expressed and purified from E. coli, and their free and immobilized forms were used to assay the degradation of alginate. The authors found these two enzymes worked well together to degrade alginate to 4-deoxy-L-erythro-5-hexoseulose uronic acid (DEH), a biofuel precursor.

Alginate is a huge feedstock in the ocean and can be transformed into biofuel molecules. The degraded product of Alginate, DEH, is easier to be converted using the industrial process because of its less viscosity in water solution than alginate. The enzymes characterized could be helpful for alginate degradation to produce biofuels. The authors should address some minor points before the acceptance.

  1. As AlyFRA and AlyFRB are endo- and exo- lyases and give different product profiles, the authors should add the background of different types of lyases in the introduction section at Line 40.
  2. Line 70 to 71. Ni affinity chromatography was used to purify the his-tagged AlyFRA and AlyFRB but not to remove the tag itself, based on the content in the Method section.
  3. Did the author characterize the products in the reactions using AlyFRA only or AlyFRB only, using LC-MS? What are the products in these reactions?
  4. The authors conducted the reactions at pH7.5 at 25oC. Did the authors optimize the reaction conditions regarding different pH and temperature?

Author Response

We are grateful to reviewer #3 for their comments and useful suggestions that have helped us to improve our paper considerably. As indicated in the responses that follow, we have taken all these comments and suggestion into account while revising the manuscript.

  1. From line 40, not only AlyFRA and AlyFRB, but also alginate-utilizing microorganisms express endo- and exo-type alginate lyases, degrading alginate to unsaturated uronic acid monosaccharides. Subsequently, its pyranose ring is enzymatically or non-enzymatically cleaved to produce 4-deoxy-L-erythro-5-hexoseulose uronic acid (DEH) (Ref. 3-5). The cleavage manner of AlyFRA and AlyFRB is described on line 45.
  2. We did not remove His-tag from these because they had no effect on activity in previous study (Ref. 7). We have added the following sentences to the manuscript. Each recombinant protein was used with the His-tag attached. (Page 6, lines 204-205)
  3. When only AlyFRB was used, the product was proven by LC/MS analysis to be DEH (Ref. 19). When only AlyFRA was used, LC/MS analysis was not performed because the products were alginate oligosaccharides (Figure 2 (a)).
  4. We did not perform the experiment at a different pH because previous study showed that the optimum pH was pH 7.5 (Ref. 19). Since the optimum temperature was 25-35°C, the experiment was conducted at 25°C to allow the enzyme reaction to take place over a longer period of time.

Round 2

Reviewer 2 Report

Thank you for the explanations. With statistical significance, it was meant to include P values and Hypothesis testing.